# Remote Estimation of Target Height from Unmanned Aerial Vehicle (UAV) Images

**Andrea Tonini** [1],*, **Paula Redweik** [2], **Marco Painho** [3] and **Mauro Castelli** [3]

1    European Maritime Safety Agency, Surveillance Unit–Remotely Piloted Aircraft Systems, Praça Europa 4, 1249-206 Lisbon, Portugal

2    Instituto Dom Luiz, Faculdade de Ciências, Universidade de Lisboa, Campo Grande, 1749-016 Lisbon, Portugal; pmredweik@fc.ul.pt

3    NOVA Information Management School (NOVA IMS), Universidade Nova de Lisboa, Campus de Campolide, 1070-312 Lisbon, Portugal; painho@novaims.unl.pt (M.P.); mcastelli@novaims.unl.pt (M.C.)

*    Correspondence: M20170214@isegi.unl.pt; Tel.: +351-961-059-638

**Abstract:** This paper focuses on how the height of a target can be swiftly estimated using images acquired by a digital camera installed into moving platforms, such as unmanned aerial vehicles (UAVs). A pinhole camera model after distortion compensation was considered for this purpose since it does not need extensive processing nor vanishing lines. The pinhole model has been extensively employed for similar purposes in past studies but mainly focusing on fixed camera installations. This study analyzes how to tailor the pinhole model for gimballed cameras mounted into UAVs, considering camera parameters and flight parameters. Moreover, it indicates a solution that foresees correcting only a few needed pixels to limit the processing overload. Finally, an extensive analysis was conducted to define the uncertainty associated with the height estimation. The results of this analysis highlighted interesting relationships between UAV-to-target relative distance, camera pose, and height uncertainty that allow practical exploitations of the proposed approach. The model was tested with real data in both controlled and uncontrolled environments, the results confirmed the suitability of the proposed method and outcomes of the uncertainty analysis. Finally, this research can open consumer UAVs to innovative applications for urban surveillance.

**Keywords:** remote surveillance; target height; UAV; pinhole model; image distortion compensation; uncertainty analysis

## 1. Introduction

Unmanned Aerial Vehicles (UAV), which have been employed for more than two decades for military activities, are nowadays widely used for civil applications as well [1]. In particular, non-coaxial multi-rotors with weight below 4 kg are often used to complement or, in some cases, even replace fixed video cameras for monitoring and surveillance activities [2,3]. In fact, UAVs can bring a very relevant added value compared with static installations: the possibility to transport and orienting the camera as needed, allowing us to perform pre-established survey paths or even follow a specific target, if needed [4]. These kinds of devices are especially used in urban areas due to their rapidness of deployment and safety, compared with larger UAVs.

Remote surveillance or monitoring activities may often require estimating the height of a target via image analysis. The target could be a tree for example, in order to monitor its growing for agricultural purposes [4], or a building, to follow construction developments, etc. However, as we may expect, remote height estimation from image analysis is very often needed to define the exact stature of human beings. This is required to support activities such as the identification of a person of interest [5].

There is a significant amount of studies in the literature dedicated to obtaining a person's body height from video footage, but, almost the totality of them considered data collected by static surveillance cameras. In some cases, vanishing lines in the scene and a reference height in the scene is required to define the height of the target (see for example [6–9]). Other authors have proposed to estimate the height of a person standing on a floor considering a pinhole camera model after camera calibration and image distortion compensation [10].

The widely used approach of using vanishing lines with a reference object present in the scene for height estimation has relevant setbacks: defining vanishing lines and reference objects may not be always possible in an image. On the other hand, the pinhole model does not require vanishing lines [11] and therefore is deemed very suitable for UAV operations [12]. Photogrammetric techniques require having either a double camera pointing at the same target or acquiring at least two images from different orientations of a (static) feature. LiDAR data needs to be acquired by devices installed in aircraft specifically designated for this kind of survey technique. In some studies related to human beings, machine learning approaches are applied to images acquired by a UAV [13].

This paper focuses on how the height of a feature standing vertically from the ground can be measured with a "regular" payload for lightweight UAVs, which is a daylight electro-optical camera installed into steerable gimbals. The goal is to estimate the height using a single image. Moreover, since in specific surveillance activities there might be a need for quick decision making during the flight, the proposed method foresees to perform the computation swiftly, to obtain results in real-time whenever needed or possible. Surveillance activities are conducted between 10/15 m until 120 m distance (to keep the visual line of sight with the UAV). However, considering that consumer-level UAV does not usually have cameras with approximation capacities (zoom), the max distance from the target considered in this study is around 50 m.

The approach foresees a correction to remove lens distortion before applying a pinhole camera model [14]. Since the correction of an entire image may be time-consuming, the approach here described would require correcting just those pixels that define the top and the bottom of the target feature. The selection of those pixels could be done automatically although, in this contribute, this was done manually to avoid possible false results. Selected pixels should be normalized to a dimensionless image, corrected using specific correction parameters and, finally, denormalized to compute the corrected pixel coordinates to be used in further calculations for height estimation. On the other hand, the image orientation requires the determination of intrinsic camera parameters, such as the focal length, and extrinsic camera parameters, such as the camera position and orientation in the object space [15].

The proposed approach does not require intensive computing because it is based on trigonometric calculations according to the pinhole model and requires correcting the distortion only for a very limited number of pixels. It is therefore deemed suitable for real-time applications if the parameters required for the calculation are also available in real-time. This paper discusses when this can be performed, taking into account the availability of required parameters (pitch angle, number of pixels panning the feature in the image plane, and camera-to-target distance) in real case scenarios.

Although performing image metrology using UAVs is not novel, this paper presents one of the very first attempts to estimate target height using the pinhole model after distortion compensation with a gimballed camera mounted into a moving platform such as a UAV. Only fixed cameras were considered in past studies. Differently to previous studies dedicated to the remote estimation of target height, where only fixed cameras were considered, this study takes into account that the position of the camera is given by positional systems (like GPS), while orientation is given by Euler angles measured by gyroscopes. The problem of retrieving the height of a target must be formulated to consider realistic UAV operational scenarios, considering that the activities may be conducted outdoor (e.g., urban areas) where the elevation and scene content is rapidly changing.

Since angles and, especially, the measurements to establish the position of the UAV (and the camera) might be affected by relevant uncertainties that may influence the correctness of the estimation,

a detailed uncertainty analysis was conducted to better explore the practical limits of the approach. Moreover, this analysis defines how the initial accuracy of the parameters required in the calculation may impact the final estimation and reinforce the validation conducted with real data.

Finally, the procedure was tested with real data collected with a regular-market lightweight quadcopter. A measuring pole of known length, standing vertically from the ground was used as a target for the acquisition of several still images taken from different positions. For each shot, the height of the target was calculated considering the proposed procedure and compared with the real height of the pole to assess the uncertainty of the estimations.

## 2. Methods

The first part of this section describes the basic principles of the pinhole model for computer vision and processes for lens distortion compensation. After that, computer vision techniques were applied to deal with cameras installed into UAVs. The last part of this section presents and describes the method to estimate the target height from still oblique images acquired with cameras installed into UAVs.

### 2.1. Pinhole Camera Model and Computer Vision

In computer vision, cameras are usually modeled with the pinhole camera model [14]. The model is inspired by the simplest camera, where the light from an object enters through a small hole (the pinhole). This model considers a central projection, using the optical center of the camera and an image plane (that is perpendicular to the camera's optical axis, see Figure 1). In the physical camera, a mirror image is formed behind the camera center but, often, the image plane is represented in front of the camera center. The pinhole camera model represents every 3D world point P (expressed by world coordinates $x_p$, $y_p$, $z_p$) by the intersection between the image plane and the camera ray line that connects the optical center with the world point P (this intersection is called the image point, noted with I in Figure 1).

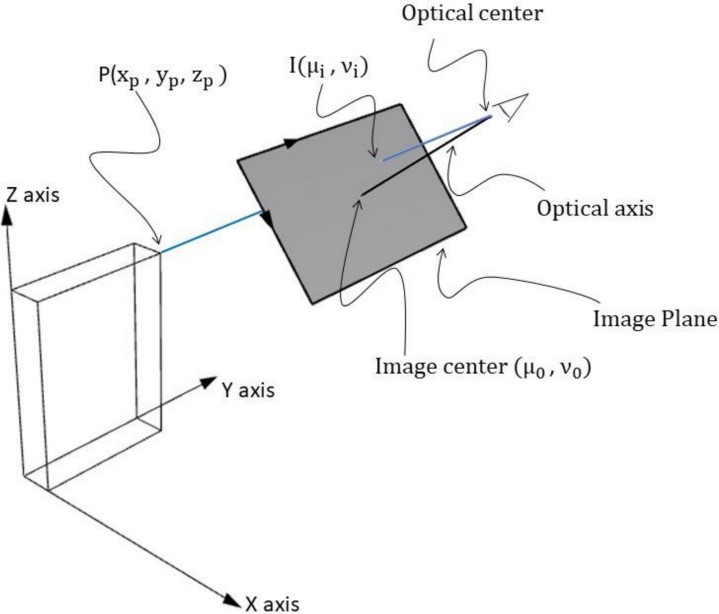

**Figure 1.** Graphical representation: a 3D world point P is projected onto a 2D image plane on point I. (Source: Personal collection, adapted from [10]).

The pinhole camera projection can be described by the following linear model [10].

$$\begin{bmatrix} \mu_i \\ v_i \\ 1 \end{bmatrix} = K[RT] \begin{bmatrix} x_P \\ y_P \\ z_P \\ 1 \end{bmatrix}. \tag{1}$$

$K$ is the calibration matrix [14], defined as follows:

$$= \begin{bmatrix} \alpha_\mu & \gamma & \mu_0 \\ 0 & \alpha_v & v_0 \\ 0 & 0 & 1 \end{bmatrix}. \tag{2}$$

$\alpha_\mu$ and $\alpha_v$ represent the focal length expressed in pixels. $\mu_0$ and $v_0$ are the coordinates of the image center expressed in pixels, with origin in the upper left corner (see Figure 1). $\gamma$ is the skew coefficient between the x and y axes, this latter parameter is very often 0.

The focal length, (which can be here considered as the distance between the image plane and the optical center) can be also expressed in metric terms (e.g., mm instead of pixels) considering the following expressions [14]:

$$F_x = a_\mu \frac{W_\mu}{w_\mu} \tag{3}$$

$$F_y = a_v \frac{W_v}{w_v}, \tag{4}$$

$w_\mu$ and $w_v$ are, respectively, the image width and length expressed in pixels, $W_\mu$ is the width and $W_v$ the length of the camera sensor expressed in world units (e.g., mm). Usually, $F_x$ and $F_y$ have the same value, although they may differ due to several reasons such as flaws in the digital camera sensor or when the lens compresses a widescreen scene into a standard-sized sensor. R and T in (1) are the rotation and translation matrices of the camera, respectively, in relation to the world coordinate system. These include the extrinsic parameters which define the so-called "camera pose". [RT] is a $3 \times 4$ matrix composed of the three columns of the rotation matrix R and the translation vector T as the fourth column. Image and object points are represented in this model in homogeneous coordinates.

R is defined in this case by the angles around the axes (X, Y and Z) of the world coordinate system needed for rotating the image coordinate system axes in order to get them coincident (or parallel) with the world coordinate system axes. In the case of rotation around the X-axis by an angle $\theta_x$, the rotation matrix $R_x$ is given by (5) [14]:

$$R_x = \begin{bmatrix} 1 & 0 & 0 \\ 0 & \cos(\theta_x) & -\sin(\theta_x) \\ 0 & \sin(\theta_x) & \cos(\theta_x) \end{bmatrix}. \tag{5}$$

Rotations by $\theta_y$ and $\theta_z$ about the Y and Z axes can be written as:

$$R_y = \begin{bmatrix} \cos(\theta_y) & 0 & \sin(\theta_y) \\ 0 & 1 & 0 \\ -\sin(\theta_y) & 0 & \cos(\theta_y) \end{bmatrix} \tag{6}$$

$$R_z = \begin{bmatrix} \cos(\theta_z) & -\sin(\theta_z) & 0 \\ \sin(\theta_z) & \cos(\theta_z) & 0 \\ 0 & 0 & 1 \end{bmatrix}. \tag{7}$$

A rotation R about any arbitrary axis can be written in terms of successive rotations about the X, Y and Z axes, using the matrix multiplication shown in (8). In this formulation $\theta_x$, $\theta_y$ and $\theta_z$ are the Euler angles.

$$R = R_z R_y R_x. \tag{8}$$

T is expressed by a 3-dimensional vector that defines the position of the camera against the origin of the world coordinate system. Scaling does not take place in the definition of the camera pose. Enlarging the focal length or sensor size would provide the scaling. The next section describes the lens distortion effects and procedures for their correction.

### 2.2. Lens Distorsion and Compensation

The pinhole model does not consider that real lenses may produce several different non-linear distortions. The major defects in cameras are the radial distortion, caused by light refraction differences along with the spherical shape of the lens. Other distortions, like the tangential distortion, which is generated when the lens is not parallel to the imaging sensor or when several component lenses are not aligned over the same axis, have minor relevance in quality objectives and will not be considered in this study. The radial distortions can usually be classified as either barrel distortions or pincushion distortions (Figure 2), which are quadratic, meaning they increase as the square of the distance from the center.

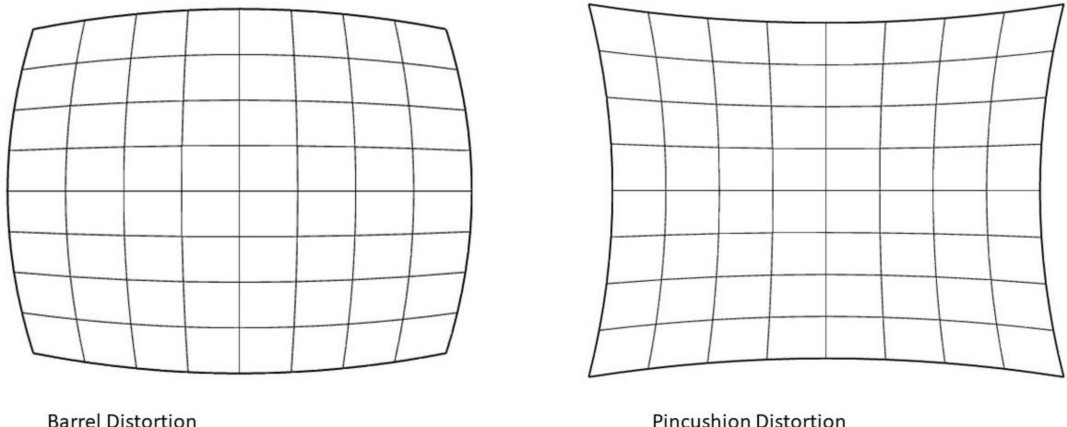

Barrel Distortion　　　　　　　　　　　　　　　　　　Pincushion Distortion

**Figure 2.** Effect of barrel and pincushion distortions (Source: Personal collection, adapted from [16]).

Removing a distortion means obtaining an undistorted image point, which can be considered as projected by an ideal pinhole camera, from a distorted image point. The simplest way to model the radial distortion is with a shift applied to the pixel coordinates [16]. The radial shift of coordinates modifies only the distance of every pixel from the image center. Let $r$ represent the observed distance (distorted image coordinates from the center) and $r_{corr}$ the distance of the undistorted image coordinates from the center. The observed distance for a point I ($\mu_i$ $\nu_i$), in the image (see Figure 1) can be calculated as follows [16]:

$$r = \sqrt{(\mu_i - \mu_0)^2 + (\nu_i - \nu_0)^2}. \tag{9}$$

With these notations, the function that can be used to remove lens distortion is [16]:

$$r_{corr} = f(r) \tag{10}$$

However, before applying the compensation function $f(r)$ we need to underline that the model would be useless if images with the same distortion, but different resolutions would have different distortion parameters. Therefore, all pixels should be normalized to a dimensionless frame, so that the

image resolution is not important. In the dimensionless frame, the diagonal radius of the image is always 1, and the lens center is (0; 0) [16].

The formula to transform the pixel coordinates to dimensionless coordinates is the following [16]:

$$
\begin{pmatrix} p_\mu \\ p_\nu \end{pmatrix} = \begin{pmatrix} (\mu_i - \mu_0)/\sqrt{\left(\frac{w_\mu}{2}\right)^2 + \left(\frac{w_\nu}{2}\right)^2} \\ (\nu_i - \nu_0)/\sqrt{\left(\frac{w_\mu}{2}\right)^2 + \left(\frac{w_\nu}{2}\right)^2} \end{pmatrix},
\tag{11}
$$

$p_\mu$ and $p_\nu$ are the dimensionless pixel coordinates and $w_\mu$, $w_\nu$ are the image width and height in pixels.

The dimensionless coordinates defined in (11) can be used to calculate a normalized distance $r_p$ considering the formula given in (9). $r_p$ can be then used to approximate the normalized $r_{corr}$ with its Taylor expansion [16]:

$$
r_{corr} = r_p + \kappa_1 r_p{}^3 + \kappa_2 r_p{}^5 + \kappa_3 r_p{}^7,
\tag{12}
$$

$\kappa_i$ are the radial distortion coefficients. The "perfect" approximation would be a polynomial of infinite degree; however for the purpose of this project, because the target is always centered in the image center (see Section 2.5) the radial distortion does not affect the measured pixels as much as if the measurements would be made at longer radial distances, as Figure 2 illustrates. Therefore, considering just $\kappa_1$ in (12) is deemed enough for this analysis. $r_{corr}$ calculated with (12). needs to be denormalized to obtain the undistorted $\mu_{i-corr}$ and $\nu_{i-corr}$ image coordinates for the image under study.

## 2.3. Elements to Consider When Dealing with Cameras Installed into UAVs Operating Outdoor

Several elements need to be taken into due consideration when operating outdoor with cameras installed into UAVs:

- The camera is usually fitted into steerable gimbals, which may have the freedom to move around one, two, or even three axes (which would be formalistically called one-gimbal, two-gimbal or three-gimbal configurations, [1]). In those cases where the gimbal has limited degrees of freedom, further steering capacity for the camera must be provided by the UAV itself via flight rotations.

- The parameters required for the transformation from world coordinate system to camera coordinate system (extrinsic parameters) can be obtained from positioning measurements (latitude, longitude, and elevation) and Euler angles (yaw, pitch, and roll). Regular GPS receivers, which are not subject to enhancements such as differential positioning, may be affected by a relevant positional error, especially in elevation. On the other hand, the orientation angles are measured by sensitive gyroscopes, gyroscopes usually have good relative accuracy, although the absolute instantaneous orientation angles may be affected by an accumulated error due to drift unless adequately compensated. Positioning systems, gyroscopes and accelerometers are used by the Inertial Measurement Unit (IMU) which are essential components for the guidance and control of UAVs [1].

- The intrinsic parameters must be known. For those cases where the UAV camera specs are not available, the intrinsic parameters (image principal point, focal length, and skew and distortion coefficients) can be retrieved using a calibration procedure [17] provided, for example, by computer vision open libraries such as OpenCV [18]. Intrinsic parameters can also be obtained from a bundle adjustment of regular photo coverage with highly convergent images [19].

- In UAV surveillance activities, the camera is usually centered and kept over the target of interest. Nowadays, this is normally performed automatically by detection and tracking algorithms [18]. Moreover, when UAV's gimbal is fitted with a laser range finder, the distances are measured on the center of the image (these devices are normally aligned to the camera center ray line). Thus, where the topography may rapidly change (like in outdoor activities), only the distance to a centered target can be correctly measured. Taking into account all these elements, the procedure

described in this paper has been developed considering a vertical target located in the center of the image.

- Each still image acquired by the UAV is usually accompanied by a set of cameras and UAV flight information, stored as metadata. The amount of information stored varies from system to system. Advanced imaging equipment may provide a complete set of metadata in Key-Length-Value (KLV) format in accordance with Motion Imagery Standards Board (MISB) standards [20]. Lightweight UAV available in the regular market are not always fitted with such advanced devices but, very often, are capable to store a minimum set of metadata, which includes on-board positioning coordinates, flight orientation and camera orientation.
- Advanced UAV imaging systems are also fitted with laser range finders, which are capable of measuring the instantaneous camera-to-target distance and store this information as metadata.

The following section describes in detail the application of the pinhole model for computer vision analysis and its parameters.

### 2.4. On-Board Sensor and Data Analysis for Height Estimation

Large UAVs, which are also called MALEs (Medium Altitude Long Endurance [21]), are usually fitted with three-gimbaled advanced imaging systems and accurate positioning systems resorting also to differential positioning techniques for higher accuracy. These systems are calibrated and capable to perform transformations in real-time and embed the instantaneous camera pose, and other information (such as Field of View (FOV), image footprint projected on the ground, and measured slant range (when available)) into the acquired video stream using the KLV encoding protocols, in accordance to military standards [22].

On the other hand, non-military lightweight UAVs available in the regular market are not always fitted with advanced imaging systems and very accurate positioning systems. For example, the DJI Phantom 4 PRO (a widely diffused multi-rotor platform of 1.388 kg, used to collect data for the testing of the approach described in this paper, see Section 3. Results) is not capable to generate KLV embedded metadata but it can generate ancillary tags in Exchangeable Image File Format (EXIF) of still images which provide, among other information, the position of the aircraft, aircraft orientation and camera orientation at the moment of the acquisition of the still image. DJI Phantom 4 PRO has a GPS/GLONASS positioning system [23]. The actual uncertainty of this positioning system after refinement with accelerometric info provided by the IMU is ±0.5 m vertically and ±1.5 m horizontally [23]. The camera of this UAV has a pivoted support (one-gimbal) with a single degree of freedom around the *Y* axis (pitch angle, see Figure 3). Angular values are measured with an uncertainty of ±0.02° [23]. However, this angular accuracy refers to the gimbal zero reference. Thus, in real case scenarios, the uncertainty associated with Euler angles could be slightly higher. Although not specified in any available technical documentation, considering the available information of this UAV, it is here assumed that the transformations employed to provide the information in the EXIF tags are the following: (a) the translation defined by the positioning coordinates of the UAV body, (b) rotation based on Euler angles of the body followed by (c) a 1D rotation of the camera (pitch angle). Therefore, the position of the camera when dealing with DJI Phantom 4 PRO can be defined by UAV body positional location (coordinates) while the orientation is given by a yaw angle defined by flight orientation, a pitch angle defined by camera orientation and a roll angle defined by flight orientation.

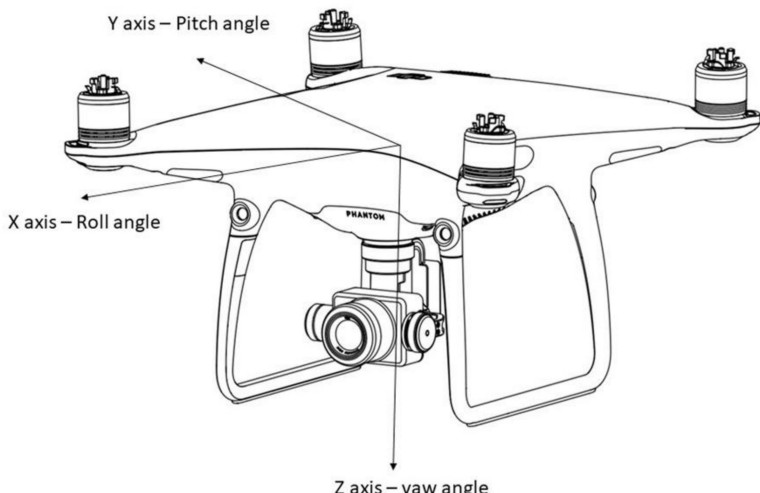

**Figure 3.** Axis and Euler angles for the case of DJI Phantom 4 PRO (Source: Personal collection, adapted from [23]).

The camera sensor is a 1″ Complementary Metal-Oxide Semiconductor (CMOS) of 20 MP effective pixels with 5472 × 3648 pixels resolution and 13.2 mm × 8.8 mm size, lens focal length of 8.8 mm with no optical zoom and diagonal FOV of 84° [23].

Assuming a lightweight UAV, like the one described in Figure 3, a feature standing vertically on the ground, for example, a pole, that the UAV has a heading (yaw angle) and pitch angle appropriate to point to the target and a Roll angle equal to zero, the situation graphically represented in Figure 4 occurs.

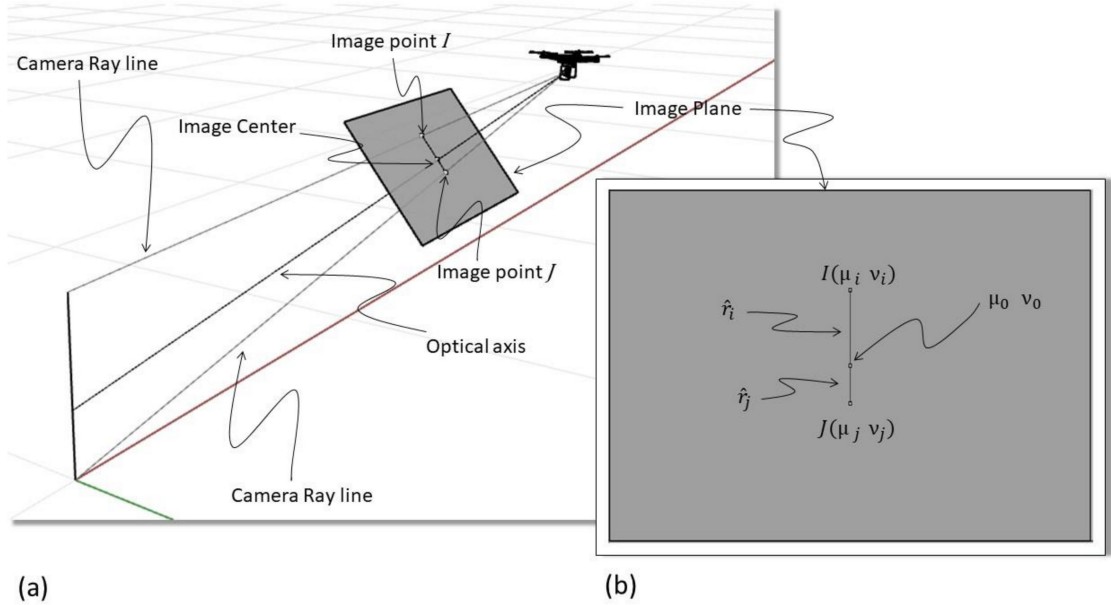

(a)                                                                                  (b)

**Figure 4.** Graphical representation of a lightweight unmanned aerial vehicle (UAV) pointing to a vertical pole (**a**) with a roll angle equal to zero. In (**b**) the image plane is represented in orthogonal view (as it would appear on a screen) (source: personal collection). Please note that (**a**) is a perspective view, please refer to (**b**) to visualize the image plane without distortion.

Point $\mu_0, \nu_0$ in Figure 4b is the image center, which is obtained, as already described, by the intersection between the image plane and the optical axis (see Figure 4a). The optical axis is centered on the target, not necessarily to the midpoint but to any point of the pole. The Image Point I is given by the intersection of the camera ray line that connects the tip (highest point) of the pole with the

camera center. This point is expressed by the image coordinates $\mu_i$, $\nu_i$ while $\hat{r}_i$ represents the distance from the image center, more specifically, to the principal point. Moreover, $\hat{r}_i$ is a distorted value that needs to be compensated to obtain the distance $r_{i-corr}$ of the ideal undistorted image. The procedure to obtain such undistorted distance was already discussed in the previous paragraph (see (12)). Similarly, Image Point J is the intersection of the image plane with the ray line that connects the bottom of the pole (lowest point) with the camera center. The point is expressed by the image coordinates $\mu_j$, $\nu_j$ while $\hat{r}_j$ represents the distance from the image center that needs to be compensated to get $r_{j-corr}$, the undistorted distance from the center of the ideal undistorted image. The line I–J in the image plane is the projected height of the pole expressed in pixels in the image plane. Considering now the case when the roll angle is different than zero graphically represented in Figure 5.

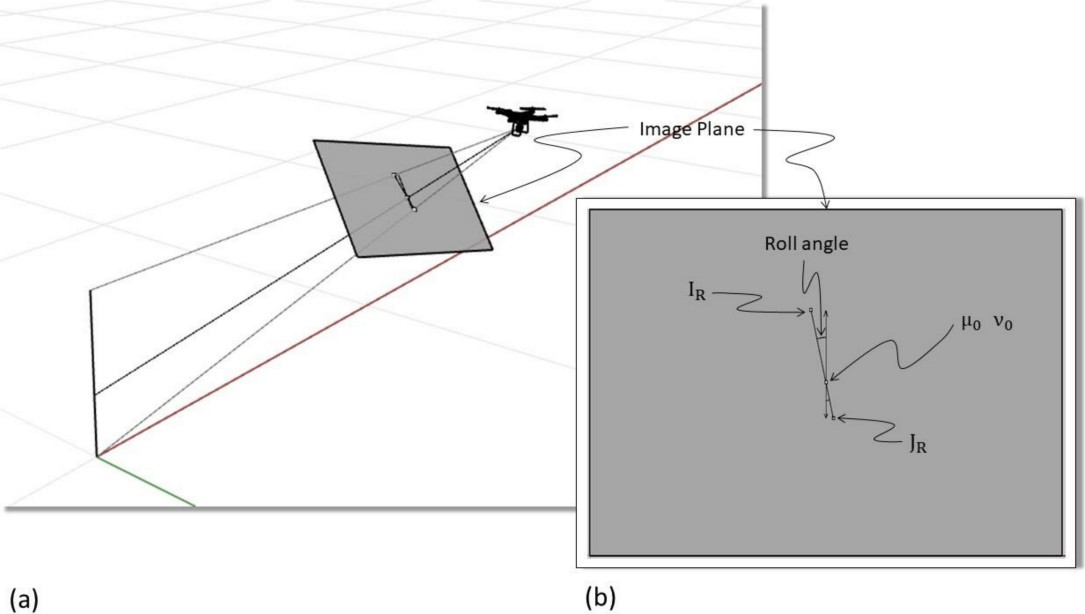

(a)                      (b)

**Figure 5.** Graphical representation of a lightweight UAV pointing to a vertical pole (**a**) with roll angle different than zero. Orthogonal view of the image plane (**b**) with the representation of the pole and indication of the roll angle (source: personal collection). Please note that (**a**) is a perspective view, please refer to (**b**) to visualize the image plane without distortion.

When the roll angle is different than zero, the line $I_R$-$J_R$, which is the representation of the pole in the image plane, will not appear as parallel to the $\nu$ axis, as in the case before, but rotated by an angle equal to the Roll angle itself, as it is possible to infer from (5). As mentioned above, the observed distances (respectively $\hat{r}_{Ri}$ and $\hat{r}_{Rj}$) must be compensated to obtain the distances $r_{Ri\ -corr}$ and $r_{Rj-corr}$ of the ideal undistorted image.

The method assumes that the distances $\hat{r}_{Ri}$ and $\hat{r}_{Rj}$ are input data, that can be measured directly on the image or derived from an automatic point detection algorithm yielding the image coordinates of the endpoints of the pole. Direct measuring will have the drawback of the need for postprocessing, contradicting the aim of near real-time results, but it will be more precise. The automatic way has always the disadvantage of retrieving false results, especially when the contrast between object and background is not well defined. The same can be said when the object does not have sharp contours. For testing the uncertainty of the method, we decided to opt for the more reliable direct measuring, but in the future, an adequate automatic algorithm shall be developed/adopted in order to automatize the workflow as far as possible.

The next paragraph describes how to estimate the height of a target standing vertically (pole) considering the elements discussed so far in this paper. As an example, we can use a lightweight UAV

like the DJI Phantom 4 PRO but the approach can be extended to any imaging system installed in steerable moving platforms.

## 2.5. Estimating Target Height with a Camera Fitted into UAVs

The approach proposed in this study for the estimation of target height using a camera fitted into UAVs foresees the UAV pointing at the target, as depicted in Figures 2 and 3, starting with the case when the roll is zero (see Figure 6).

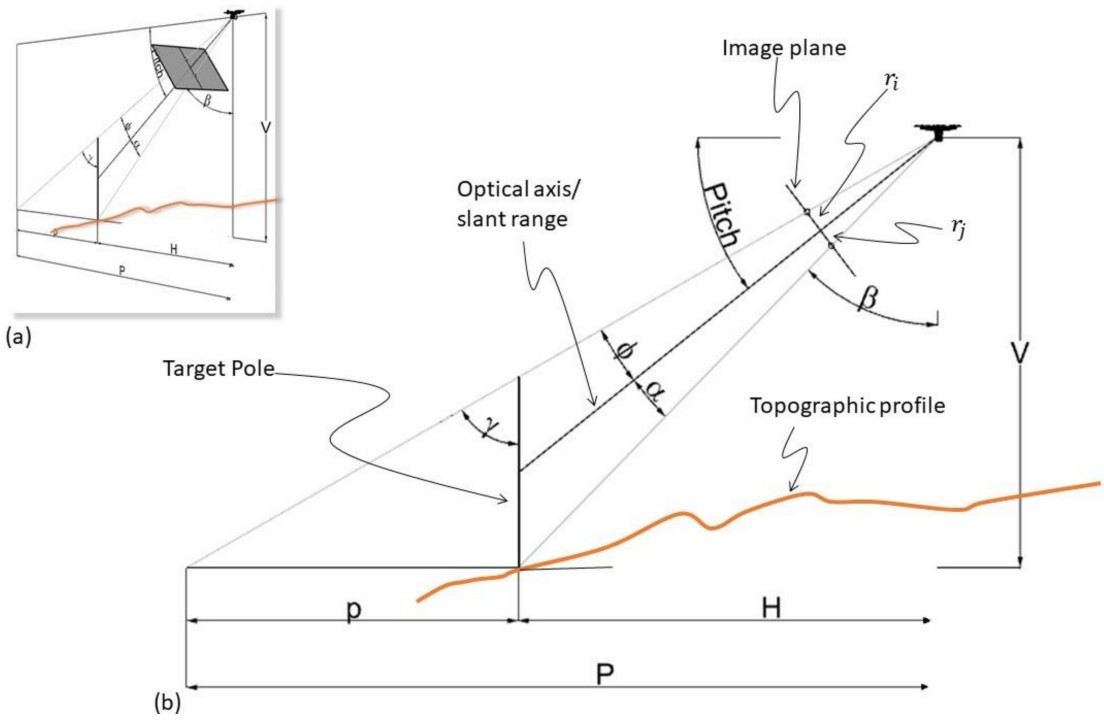

**Figure 6.** Perspective view of a lightweight UAV pointing to a vertical target (pole) (**a**). Orthogonal view of the same scene with descriptions (**b**) (source: personal collection).

The pitch angle, which can be also identified with $\theta_y$, see (7) is a known value, while the angles $\alpha$, $\beta$, $\phi$, $\gamma$ are not originally known but they can be retrieved using simple trigonometric calculations:

$$\alpha = \tan^{-1}\left(\frac{r_{j-corr}}{F}\right) \tag{13}$$

$$\beta = 90 - \left(\theta_y + \alpha\right) \tag{14}$$

$$\phi = \tan^{-1}\left(\frac{r_{i-corr}}{F}\right) \tag{15}$$

$$\gamma = (\phi + \beta + \alpha), \tag{16}$$

where $r_{i-corr}$ can be calculated considering (12). in the previous paragraph starting from the observed $r_i$ in the image plane (see Figure 6). Similarly, $r_{j-corr}$ refers to the point $J$ (see also Figure 4). $F$ is the focal length, which was defined by (3) and (4) (assumed here for simplicity that $F = F_v = F_\mu$).

V in Figure 6 is the vertical distance between the base of the target and the camera center, while H is the horizontal distance between the target and the camera center. H and V are not related at all to the topography, as is possible to infer from Figure 6 because H and V depend on the relative height and horizontal distance between the UAV and the target foot. If the coordinates of the target are known, then H and V are also known since the positioning coordinates of the camera are available (see Section 2.3.

Cameras installed into UAVs). The positional uncertainty and how it will impact the estimation of the target height will be treated later in this paper, but, since it is much more practical to handle horizontal distances, using a GIS (Geographic Information System (GIS), for example, than vertical elevations and because we want to reduce the number of parameters for the model (possible source of error), V is always calculated in function of H, as defined in (17) below.

$$V = H \tan(90 - \beta). \tag{17}$$

The angles $\alpha$, $\beta$, $\phi$, $\gamma$ are now known, as well as the pitch angle, $V$ and $H$. These elements can be used to calculate the height of the target using triangle similarity. In fact, $P$ (see Figure 6) can be calculated as follow:

$$P = V \tan(\alpha + \beta + \phi), \tag{18}$$

$p$ is the horizontal distance between the base of the target and the camera ray that passes through the tip (highest point) of the target, which can be calculated as follow:

$$p = P - H. \tag{19}$$

Finally, the height of the target can be calculated by (20):

$$Height\ of\ the\ Target = p \tan(90 - \gamma). \tag{20}$$

As already mentioned, the horizontal distance between the target and the camera center can be determined if the coordinates of the target are known. In practice, this could be the case only when dealing with immobile features like light poles, trees or buildings. If the position of the target is not known, as it may happen for moving targets like humans, vehicles, etc., the best solution would be using a device (like a laser range finder) to measure the instantaneous camera-to-target distance (slant range) [24], as required by dedicated military standards for UAV metadata sets [25]. As already mentioned, advanced imaging systems are very often fitted with such devices, and the instantaneous distance measurements can be stored in the KLV metadata set [21] and transferred in real-time to the ground controller [22].

Slant range is a distance is aligned with the optical axis of the camera (see Figure 6) and can be used to calculate the horizontal distance $H$ using the following formula:

$$H = Slant\ Range\ Distance * \sin(90 - Pitch\ angle). \tag{21}$$

It is important to highlight that also the slant range distances measured by laser range finders are affected by a certain error that must be considered during the estimation of the target height. For example, an error of ±0.3 m along the slant range when the horizontal camera-to-target distance is 20 m and the pitch angle is 30° will result in a horizontal error of ±0.26 m.

Let us now analyze the case in which the roll angle is different from zero: in this case, as already discussed (see Figure 5), the points I and J are not located along the $v$ axis passing on the center of the image. In other words, a vertical feature will appear as "tilted" in the image on an angle equal to roll. However, as it is possible to infer from (8) and as graphically represented in Figure 7, I and J are in the same (vertical) plane of the pitch angle. Thus, the approach presented in this paper does not need to consider the roll angle for the calculation of the target height. In this case, it is necessary to perform a distortion correction to obtain $r_{i-corr}$ and $r_{j-corr}$ and use these parameters in the formulas previously described (see (13)and (15)).

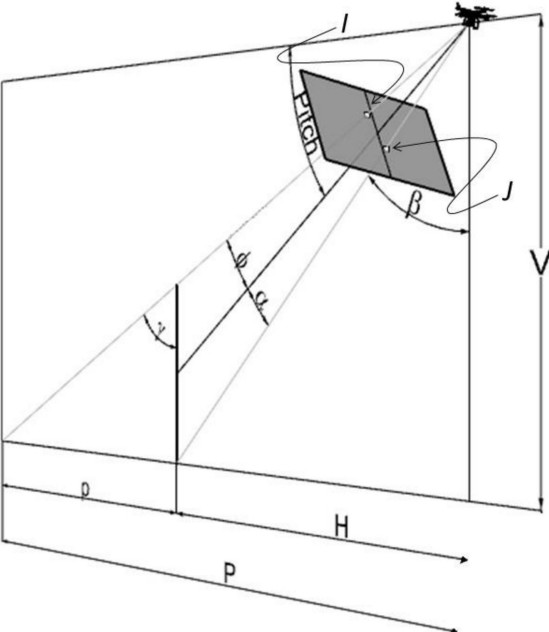

**Figure 7.** Perspective view of the image plane with visualization of the I and J, representing respectively the top and the bottom of the pole in the image plane (source: personal collection).

As previously discussed, the camera-to-target horizontal distance (H) is a required parameter in the calculations. If the target is moving (like a person), a laser range finder should be preferably used because it can measure the instantaneous slant range, from which we can derive H (the horizontal distance between the camera and the target). These devices are usually fitted only in larger UAVs, but we believe it is important to analyze this option in our paper. Regular-market UAVs (like DJI Phantom used in the field test of the paper) do not normally have such devices, but H can be calculated considering the coordinates of the target and the coordinates of the UAV. If the target is a fixed feature, like an electrical pole or a tree, then it is most probably visible in reference ortho-photo maps or topographic maps, where it is possible to retrieve its coordinates (even public available geo-webservices such as Google Earth can be used for this purpose). If the target is not fixed (therefore moving, like a human), then it is necessary to find its coordinates when the image was taken. This is not an easy task, but practical experience has shown that, especially in urban areas, the image can be taken when the target is located over a recognizable marker on the ground (like road markings) or feature (like a building corner). The coordinates of this marker or feature, which can be once again retrieved from available maps or geo web-services, can be used to indirectly obtain the horizontal target-to-camera distance.

### 2.6. Uncertainty Analysis

Assuming an accurate correction of image distortion, the relevant sources of error may come from: the horizontal camera-to-target distance, the number of pixels spanning the target from the image center toward the top and/or bottom, and angular measures.

The Computer-Aided Design (CAD) software Rhinoceros [26] was combined with a spreadsheet to visually analyze how the aforementioned uncertainties impact the height estimation. In this CAD, it was possible to create a 3D environment of real planar coordinates and locate in there a UAV (e.g., a DJI Phantom 4 PRO) in a desired location and altitude, rotate the camera of the UAV considering Euler angles (e.g., 30° pitch angle), create a target of defined height (e.g., a pole of 1.80 m), and locate it to a desired distance from the UAV. Moreover, since the pinhole model after distortion correction is perfectly geometrical, it was also possible to recreate the geometry of the camera (field of view, image plane, etc.). In this way, we can have all the parameters under control, including the number of pixels spanning the feature, to simulate different scenarios to analyze the uncertainty.

Firstly, let us consider the following scenario: a DJI Phantom 4 PRO is looking at a target of 1.80 m, the laser range finder (this UAV does not have a laser range finder, but we assume it would have one) measured a slant range distance of 23.1 m from the target (equal to 20 m horizontally), the total number of pixels spanning the feature for this case would be 248 (see Figure 8a). If the laser range finder has an uncertainty of ±0.30 m (±0.26 m horizontally), the real horizontal distance of the UAV to the target must be included between 19.74 m and 20.26 m (see Figure 8b,c). In this case, we could say that the error associated with the estimated height is ±0.02 m (varying from 1.78 m to 1.82 m as shown in Figure 8b,c). Moreover, in this case, the difference in elevation between the UAV and the target is not known because we do not have the coordinates of the target. Let us now consider the other possible source of error: the number of pixels is miscounted (this can be ideally indicated as error of collimation, where, basically a camera ray line is not collimating to the top or bottom of the target). Practical experience has shown that there might be an error of uncertainty of ±3 pixels considering a simple linear vertical feature like a pole, this uncertainty takes place at the top and at the bottom of the feature. If the target is more "complex" than a pole (like a person) this uncertainty could be more than 3 pixels. Considering the same situation described above (DJI Phantom 4 PRO looking at a target of 1.80 m, 20 m horizontal distance, 30 deg Pitch), an uncertainty of ±3 pixels at the top generates an error in the estimation of the height equal to ±0.021 m (see Figure 8d,e where the error is exaggerated to underline the issue). The same uncertainty at the bottom of the feature generates an error of 0.023 m. These two errors, although quite similar, should be kept separate because they may vary significantly for pitch assuming high values.

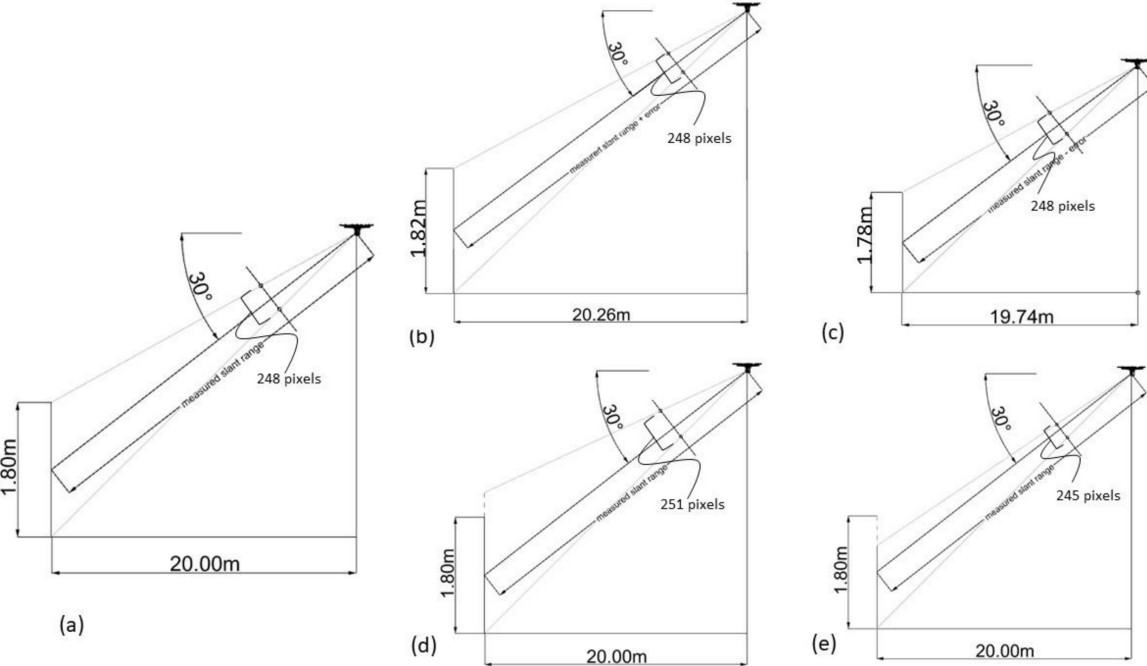

**Figure 8.** Case scenario developed in the 3D CAD Rhinoceros where a DJI Phantom 4 PRO is looking at a target of 1.80 m from a horizontal distance of 20 m. Slant distance is measured through a laser range finder (**a**). The error associated with the measured slant range (assumed to be ±0.30 m as an example) generates uncertainty in the horizontal distance, which lays between 20.26 m and 19.74 m. Keeping constant the other parameters (pitch angle equal to 30° and total pixels equal to 248), the horizontal uncertainty generates uncertainty in the calculation of height, which lays between 1.82 m (**b**) and 1.78 m (**c**). The camera ray line is not collimating to the top or bottom of the target: in other words, the erroneous value of the distance from the image center measured in pixels that may generate an error in the estimation of the height of the target. A 3-pixel difference may generate an error of ±0.021 m ((**d**) and (**e**), the error is exaggerated to underline the issue) (source: personal collection).

According to the statistical propagation of error, these three errors can be combined as follow:

$$\delta = \sqrt{\left(q_{real} - q_{H-err}\right)^2 + \left(q_{real} - q_{PixTop-err}\right)^2 + \left(q_{real} - q_{ixBot-err}\right)^2}, \quad (22)$$

where $\delta$ denotes the error, $q_{real}$ is the real height of the target (in this case 1.800 m), $q_{H-err}$ is the calculated height of the target with uncertainty related to horizontal distance (1.820 m), $q_{PixTop-err}$ is the calculated height of the target with uncertainty related to the number of pixels toward top (1.821 m), $q_{PixBot-err}$ is the calculated height of the target with uncertainty related to the number of pixels toward the bottom (1.823 m). This analysis tells us that, statistically, we should expect an error of ±0.040 m.

The analysis conducted above does not consider the uncertainty associated with angular measures. The declared angular uncertainty of the Phantom 4 PRO is ±0.02° (see Section 2.4) but, as already explained, this angular accuracy refers to the gimbal zero references. Thus, in real case scenarios, it is necessary to consider possible uncertainty. In this analysis, an angular uncertainty of ±1° is assumed for the Pitch angle. In the case here under consideration, the H value is calculated considering the Pitch Angle and the slant range (see (21)). There is a direct dependency between these three variables. According to the error propagation law, the uncertainty of two multiplied quantities is equal to the addition of their respective uncertainties (slant range: ±0.30 m and Pitch angle: ±1°). This resulting uncertainty associated with the horizontal distance H can be then used to replace $q_{H-err}$ in (22). The uncertainty of H given by the combination of slant range uncertainty and angular uncertainty is ±0.32 m which generates, statistically, an error of ±0.05 m when combined with the other uncertainties (PixTop and PixBot) in Equation (22). Once again, this is valid for pitch equal to 30° and H equal to 20 m, which is the most probable scenario for the purpose of this project. In other conditions, the error will be clearly different and therefore should be recalculated.

Let us now consider a more realistic scenario, by taking into account consumer-level UAVs, such as DJI Phantom 4 PRO, which is usually not fitted with laser range finders. In this case, we need to take into account the positional uncertainty of the UAV (the declared horizontal uncertainty of the Phantom 4 PRO is ±1.50 m [26]), the precision of the maps used to get the coordinates of the target (for a map at 1:4000 scale, we may consider uncertainty of ±1 m [27]), the uncertainty related to the number of pixels at the top and bottom, and ±1° for the Pitch, as seen before. The statistical error generated by the combination of these uncertainties can be calculated considering (22). To better analyze how the error may vary in function of the relative position of UAV with respect to the target, 20 cases with a random position of the UAV (from 5 m to 50 m horizontal distance from the target) and camera pitch angle (between 5° to 75°) were generated in a spreadsheet and visualized in a 3D CAD. These cases (artificially created) gave us the possibility to have full control over all the parameters in order to isolate and analyze each uncertainty and to calculate their combination ($\delta$). The results of this analysis are reported in Table 1.

**Table 1.** The 20 cases with a random position of the UAV generated artificially. The table reports the parameters (pixels upwards and downwards for image center, pitch angle and horizontal camera-to-target distance) with a target of 1.80 m. The table also reports the calculated height of the target when uncertainties such as UAV positional uncertainty (1.5 m), target position (1.0 m) and the number of pixels at top and bottom (3 pixels for both). Each uncertainty was isolated and treated separately to calculate the resulting height. Finally, the table reports also the statistical error obtained using the formula in (22).

|  | 1 | 2 | 3 | 4 | 5 | 6 | 7 | 8 | 9 | 10 | 11 | 12 | 13 | 14 | 15 | 16 | 17 | 18 | 19 | 20 |
|---|---|---|---|---|---|---|---|---|---|---|---|---|---|---|---|---|---|---|---|---|
| Pixels upwards ($r_i$) | 285 | 358 | 206 | 266 | 70 | 122 | 100 | 51 | 113 | 91 | 67 | 95 | 117 | 174 | 149 | 222 | 146 | 85 | 51 | 41 |
| Pixels downwards ($r_j$) | 508 | 132 | 224.3 | 101.2 | 136.8 | 55.9 | 51.3 | 139.50 | 48 | 42.9 | 46.1 | 57.8 | 193 | 524 | 419 | 137 | 92 | 77 | 87 | 28 |
| Pitch angle (degrees) | 36.8 | 49.7 | 30.5 | 35.3 | 14.1 | 27.8 | 31.7 | 6.2 | 10.9 | 17.0 | 23.2 | 41.5 | 27.4 | 19.5 | 11.4 | 49.9 | 37.2 | 37.2 | 63.7 | 72.8 |
| Horizontal distance (H) (m) | 5.0 | 6.0 | 11.3 | 12.3 | 29.7 | 29.2 | 31.7 | 33.9 | 39.7 | 45.0 | 49.1 | 24.3 | 16.5 | 8.1 | 10.9 | 7.7 | 17.6 | 25.4 | 9.1 | 8.3 |
| Real Target Height (m) | 1.80 | 1.80 | 1.80 | 1.80 | 1.80 | 1.80 | 1.80 | 1.80 | 1.80 | 1.80 | 1.80 | 1.80 | 1.80 | 1.80 | 1.80 | 1.80 | 1.80 | 1.80 | 1.80 | 1.80 |
| Height Estimation considering uncertainties |  |  |  |  |  |  |  |  |  |  |  |  |  |  |  |  |  |  |  |  |
| UAV pos. Unc. (m) | 2.34 | 2.25 | 2.04 | 2.02 | 1.89 | 1.90 | 1.88 | 1.88 | 1.87 | 1.86 | 1.86 | 1.91 | 1.96 | 2.14 | 2.04 | 2.15 | 1.96 | 1.91 | 2.09 | 2.13 |
| Target pos. Unc. (m) | 2.16 | 2.10 | 1.96 | 1.94 | 1.86 | 1.86 | 1.85 | 1.85 | 1.84 | 1.84 | 1.84 | 1.87 | 1.91 | 2.03 | 1.96 | 2.03 | 1.91 | 1.87 | 1.99 | 2.02 |
| Num. Pix. Top Unc. (m) | 1.81 | 1.81 | 1.81 | 1.81 | 1.82 | 1.83 | 1.83 | 1.83 | 1.83 | 1.84 | 1.85 | 1.83 | 1.82 | 1.81 | 1.80 | 1.81 | 1.82 | 1.83 | 1.83 | 1.88 |
| Num. Pix. Bottom Unc. (m) | 1.81 | 1.81 | 1.81 | 1.81 | 1.83 | 1.83 | 1.83 | 1.83 | 1.83 | 1.84 | 1.85 | 1.84 | 1.82 | 1.81 | 1.81 | 1.82 | 1.83 | 1.83 | 1.84 | 1.89 |
| Angular (°) | 0.05 | 0.07 | 0.04 | 0.04 | 0.02 | 0.04 | 0.04 | 0.01 | 0.01 | 0.02 | 0.03 | 0.06 | 0.05 | 0.03 | 0.01 | 0.08 | 0.05 | 0.05 | 0.13 | 0.23 |
| Statistical Error | 0.65 | 0.55 | 0.29 | 0.26 | 0.11 | 0.13 | 0.11 | 0.10 | 0.09 | 0.09 | 0.10 | 0.15 | 0.20 | 0.41 | 0.29 | 0.43 | 0.20 | 0.14 | 0.38 | 0.47 |

It is interesting to analyze the appearance of the variables (horizontal camera-to-target distance, number of pixels towards top/bottom, and pitch angle) plotted against the correspondent errors generated for the calculation of the height (Figure 9). Since the horizontal camera-to-target distance has two uncertainties associated (UAV positional accuracy and target positional accuracy), this variable was plotted against those two errors (Figure 9a,b). In both cases, the relationship can be very well correlated using a negative power function, which tells us that the error is lower when the horizontal distance is higher. This should not be surprising, because looking at the geometry in Figure 4 (and associated formulas) it is evident that the same horizontal positional uncertainty of the UAV has a stronger impact (higher uncertainty in the estimation of the height) when the target is closer. Concerning the number of pixels (both top and bottom, Figure 9c,d), we can instead notice that the error generally increases when the number of pixels decreases. This is logical since the number of pixels spanning a feature decreases with the distance, and the associated error is higher when the target is farther from the UAV. The angular uncertainty of pitch generates a trend that increases exponentially when the angle is higher (Figure 9e). Let us now consider the statistical error vs. itch (Figure 9f). Although quite scattered, Figure 9f shows that the error increases when the angle also increases. Concerning the number of pixels (in this case, we considered all the pixels for simplicity) Figure 9g shows that the error increases when the number of pixels also increases. Let us now consider the overall statistical uncertainty vs. the horizontal distance (Figure 9h): a negative power function relationship can be used to approximate this distribution (in this case, the function of the regression line is $y = 2.7182x^{-0.905}$). This well-predictable behavior is probably due to the strong component of uncertainty related to the horizontal distance. The error varies from ±0.65 m when the horizontal distance between the UAV and the target is 5 m, to 0.1 m when the distance is around 50 m. Finally, we should notice that error at around 50 m (see case 11 in Table 1) is slightly higher than the previous one (case 10). This indicates that, possibly, there is growth in the error after 50 m due to the uncertainties related to the number of pixels and pitch (especially for high angles).

Theoretically, the function of the regression curve that approximates the distribution of the statistical error against H ($y = 2.7182x^{-0.905}$) could be used to predict quite accurately the uncertainty of a height estimation at any horizontal distance (since $R^2 = 0.9849$). More practically, we need to take into account the following: (a) the function is valid for a target of 1.80 m, but in a real case scenario the height of the target would be clearly not known; (b) in a real case scenario we would not have the exact horizontal distance but just a distance affected by a certain uncertainty; (c) the function is only valid for the system (DJI Phantom 4 PRO) and for the conditions considered.

Regarding the first point (a), let us consider Figure 10 where the statistical error has been calculated for ten different H considering a height of target = 1.60 m, 1.80 m and 2.0 m. A different height of the target is clearly generating a different distribution of the error but is always well-aligned along a negative power function. Moreover, the distribution flattens when the horizontal distance is above 15 m.

Regarding point (b) mentioned above, we can use the same rationale: the function of the regression curve that approximates the distribution of the statistical error against H gets rather flat for values above 15 m to 20 m. Therefore, values of H above 15 m or 20 m, even if affected by uncertainty, can be used in the equation reported in Figure 9e without introducing a sensible error. Finally, regarding point (c), we can only say that every case is different: conditions and uncertainties must be carefully considered and analyzed, because they may generate different errors.

All in all, we can say that in cases as the one described in this study (when the horizontal uncertainty is constant and brings the highest contribution in terms of uncertainty) the statistical error has a well predictable distribution (negative power). This behavior can be used to predict a priori the uncertainty using only the horizontal target-to-camera distance. However, this prediction can be only used (cautiously) when this distance is above 15 m to 20 m (assuming that the target is a person).

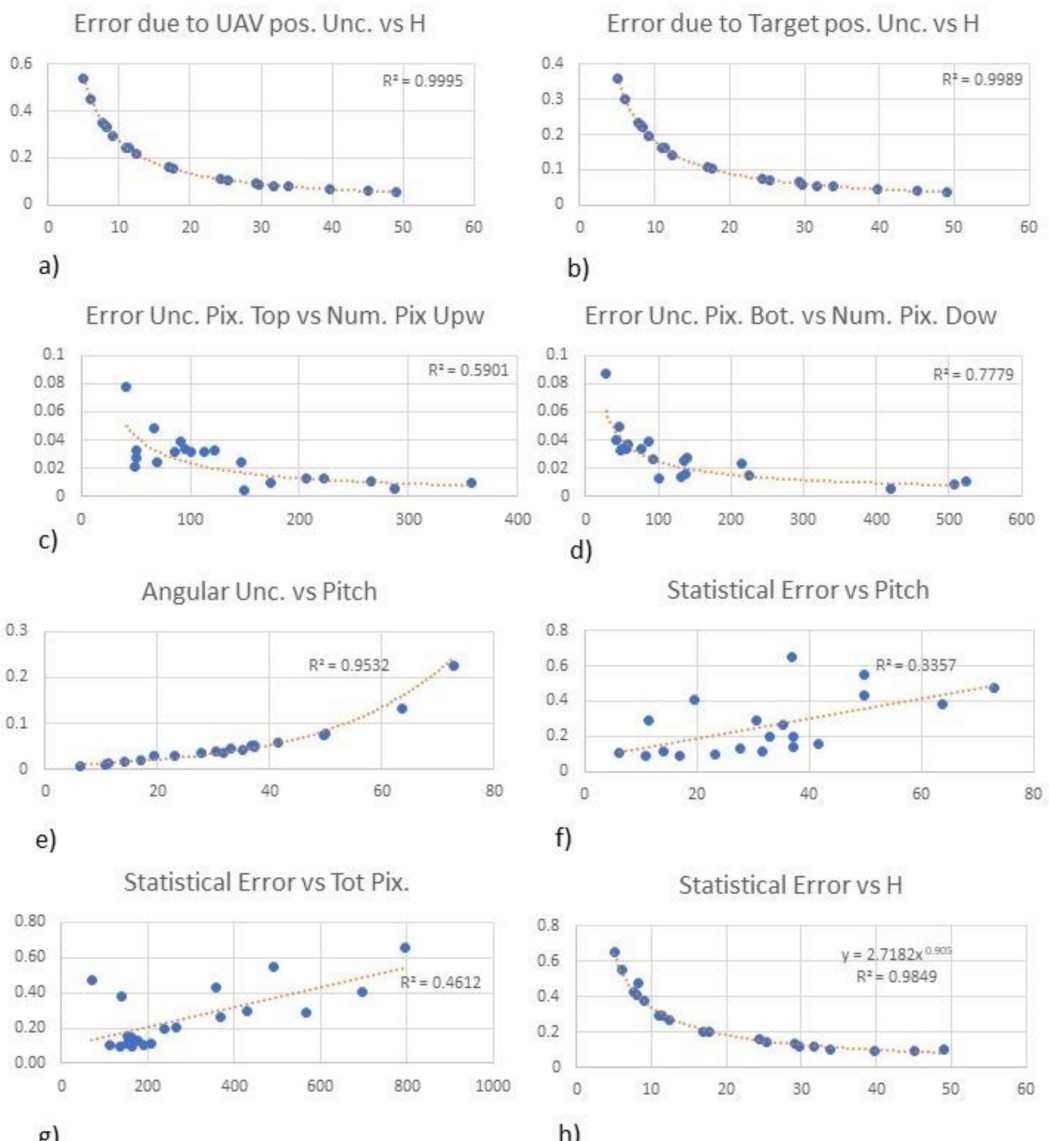

**Figure 9.** (**a**) Error in the estimation of height (obtained as: real target height—height estimation considering uav positional uncertainty) plotted against H (see Table 1). (**b**) Error in the estimation of height (obtained as: real target height—height estimation considering target positional uncertainty) plotted against H (see Table 1). (**c**) Error in the estimation of height (obtained as: real target height—height estimation considering uncertainty number of pixels toward top) plotted against pixels upwards (see Table 1). (**d**) Error in the estimation of height (obtained as: real target height—height estimation considering uncertainty number of pixels toward the bottom) plotted against Pixels downwards (see tabreftabref:remotesensing-938194-t001). (**e**) Angular uncertainty plotted against the pitch angle (see Table 1). (**f**) Combined statistical error plotted against pitch (see Table 1). (**g**) Combined statistical error plotted against the total pixels (pixels downwards + pixels upwards, see Table 1). (**h**) Combined statistical error plotted against H (see Table 1).

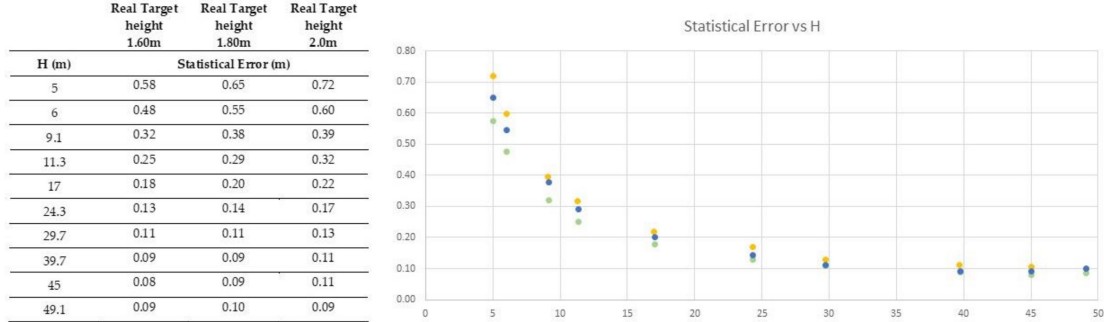

**Figure 10.** Statistical error calculated for a target of 1.60 m, 180 m and 2.0 m for ten different horizontal camera-to-target distances (on the right). Statistical error plotted against the horizontal camera-to-target distances (on the left, green circles: target height = 1.60, blue circles: target height = 1.80 m, yellow circles: target height = 2.0 m).

## 3. Results of the Field Data Tests

The procedure to estimate the target height described in the previous section was tested using real data acquired with a DJI Phantom 4 PRO (see Section 2.3. Cameras installed into UAVs for technical details regarding this device and camera used). Two separate tests were conducted: a first one realized in a controlled environment to verify the correctness of the model and a second one, conducted in an uncontrolled environment outdoor, to assess the uncertainty of this model in a real case scenario with a target of localized in a known position. DJI Phantom 4 PRO is not fitted with a laser range finder, therefore, in this study, it was not possible to test the case of moving targets.

### 3.1. Field Test 1: Validation of the Model

In this first field test, we used as a target, a wooden pole of 180 cm standing vertically from the ground. The data were acquired in a flat area (an outdoor basketball court, see Figure 11), the UAV took-off in a perfectly vertical lift from known distances from the pole (3 m, 5 m, 15 m and 20 m) measured using a metric tape. In this way, it was possible to have full control of horizontal camera-to-target distance. 10 still images were acquired with different camera pitch angles and, in each acquired image, the principal point was always oriented over the pole (an arbitrary point along the pole, as defined in Figure 5). Images not properly oriented (a principal point not located over the pole) were discarded and not used in this study. Camera Pitch angle and flight Roll angle of each image were extracted from EXIF tags, while the number of pixels spanning upward from image principal point ($r_i$) and downward ($r_j$) were measured manually on screen. Table 2 provides the details for all the acquired images.

In Section 2.2, the procedure to obtain a corrected distance from the image center was described. Such a procedure was applied to each image obtaining the $r_{i-corr}$ (number of corrected pixels from image center upward to pole's top point) and $r_{j-corr}$ (number of corrected pixels from image center downward to pole's bottom). The total number of pixels spanning the entire pole after distortion correction ($r_{i-corr} + r_{j-corr}$) as well as the calculated elevation of the UAV (defined using the formula (17)) are also reported in Table 2. The Distortion Coefficient to be used for the correction was retrieved through camera calibration techniques [15] developed with OpenCV via Python programming and is equal to 0.0275 (this is the k1 coefficient as expressed in the model (12)). Please see Section 2.2 for further clarification. Considering the good results of this test, errors in pitch and image measurement of the top and bottom of the target can be considered negligible.

Considering the small errors obtained (root mean square = 0.016 m, standard deviation = 0.015 m) probably due to an unperfect vertical lift of the UAV or angular uncertainty, the model is considered validated.

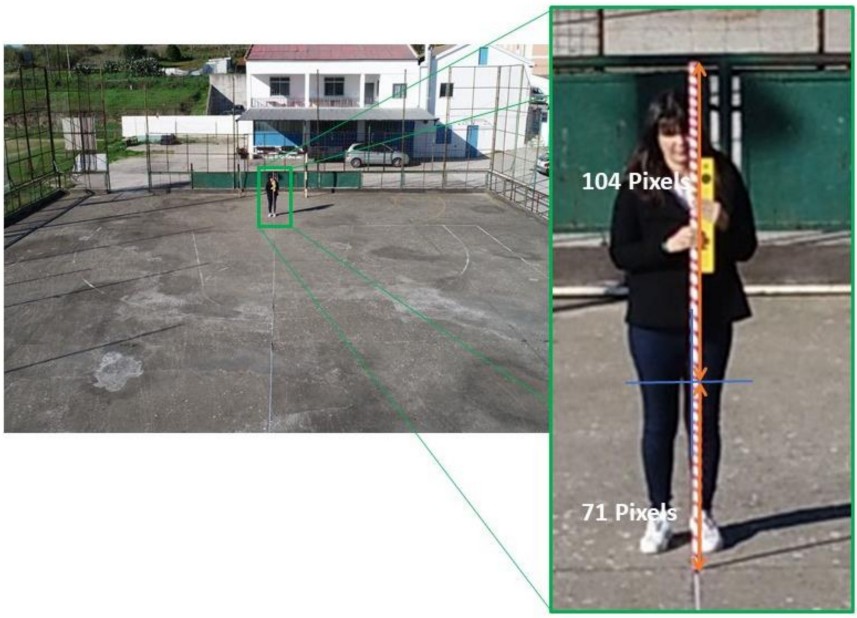

**Figure 11.** An example of a still image acquired with DJI Phantom 4 PRO with image principal point (visualized in the picture with a blue cross) located over the pole. The number of pixels spanning upward (104) and downward (71) from the image principal point were measured manually on screen. (Source: Personal collection).

**Table 2.** Data collected during Field Test 1 and results.

|  | DJI39 | DJI42 | DJI43 | DJI49 | DJI48 | DJI122 | DJI126 | DJI80 | DJI84 | DJI89 |
|---|---|---|---|---|---|---|---|---|---|---|
| Pixels upwards ($r_i$) | 306 | 272 | 158 | 284 | 358 | 179 | 288 | 220 | 108 | 104 |
| Pixels downwards ($r_j$) | 1660 | 1706 | 1548 | 510 | 760 | 226 | 30 | 107 | 157 | 71 |
| Pitch angle (deg) | 11.7 | 11.7 | 19.3 | 36.8 | 20.3 | 14.7 | 33.7 | 2.7 | 25.8 | 43.3 |
| Distance (H) (m) | 3 | 3 | 3 | 5 | 5 | 15 | 15 | 20 | 20 | 20 |
| Calculated UAV altitude (V) | 2.19 | 2.2 | 2.7 | 5.0 | 3.1 | 4.9 | 10.2 | 1.5 | 10.8 | 19.6 |
| Flight Roll Angle (deg) | 0.0 | 1.3 | 0.9 | 0.7 | 2.2 | 0.6 | 0.2 | 2.7 | 0 | 2.4 |
| Total number of pixels (after distortion correction) | 1954 | 1965 | 1697 | 794 | 1117 | 405 | 318 | 327 | 265 | 175 |
| RESULTS |  |  |  |  |  |  |  |  |  |  |
| Estimated height (m) | 1.82 | 1.84 | 1.81 | 1.79 | 1.82 | 1.79 | 1.80 | 1.79 | 1.80 | 1.80 |
| Real Height (m) | 1.80 | 1.80 | 1.80 | 1.80 | 1.80 | 1.80 | 1.80 | 1.80 | 1.80 | 1.80 |
| Error (m) | −0.02 | −0.04 | −0.01 | 0.01 | −0.02 | 0.01 | 0.00 | 0.01 | 0.00 | 0.00 |

*3.2. Field Test 2: Real Case Scenario with a Target of Known Position*

In this field test, it was used the same wooden pole of 180 cm standing vertically from the ground in a position of known coordinates (Lat, Long) retrieved in a digital topographic map at the scale 1:5000 managed in a GIS. 32 still images were acquired with different camera poses (see Tables 3 and 4) and, in each acquired image, the principal point was always oriented over the pole. All those images not properly oriented (the principal point not located over the pole) were discarded. The images were acquired in an open space with good visibility to satellites.

**Table 3.** Data and results for the first 16 still images acquired with the lightweight UAV with a Pole of 1.80 m as target.

| | DJI31 | DJI32 | DJI34 | DJI41 | DJ09 | DJI12 | DJI13 | DJI15 | DJI17 | DJI18 | DJI33 | DJI35 | DJI36 | DJI37 | DJI125 | DJI42 |
|---|---|---|---|---|---|---|---|---|---|---|---|---|---|---|---|---|
| Number of pixels upwards ($r_i$) | 355 | 680 | 172 | 151 | 1358 | 1352 | 942 | 931 | 592 | 363 | 678 | 150 | 334 | 344 | 154 | 156 |
| Number of pixels downwards ($r_j$) | 1150 | 164 | 204 | 142 | 49 | 69 | 82 | 92 | 48 | 61 | 160 | 216 | 203 | 514 | 297 | 141 |
| Gimbal pitch angle (degrees) | 13.7 | 13.7 | 5.6 | 14.8 | 38.7 | 38.7 | 26.4 | 26.4 | 16 | 10.1 | 13.7 | 12 | 17.9 | 30.6 | 26.7 | 14.8 |
| Flight roll angle (degrees) | 1.8 | 0.9 | 4.2 | 5.2 | 0.4 | 0.8 | 0.5 | 0 | 0.6 | 0.3 | 1.2 | 2.9 | 3.3 | 0.8 | 3.6 | 1.3 |
| Horizontal distance (H) (m) | 4.2 | 7.9 | 17.1 | 21.28 | 3.6 | 3.64 | 6.04 | 6.17 | 10.28 | 14.96 | 7.95 | 17.38 | 11.62 | 6.04 | 11.78 | 21.12 |
| Total number of pixels (after distortion correction) | 1501 | 843 | 376 | 293 | 1401 | 1415 | 1022 | 1021 | 639 | 424 | 837 | 366 | 537 | 858 | 451 | 297 |
| Est. height (m) | 1.94 | 1.87 | 1.78 | 1.83 | 1.77 | 1.82 | 1.89 | 1.93 | 1.87 | 1.77 | 1.87 | 1.83 | 1.87 | 1.98 | 1.87 | 1.84 |
| Error (m) | 0.14 | 0.07 | −0.02 | 0.03 | −0.03 | 0.02 | 0.09 | 0.13 | 0.07 | −0.03 | 0.07 | 0.03 | 0.07 | 0.18 | 0.07 | 0.04 |

**Table 4.** Data and results for the remaining 16 still images acquired with the lightweight UAV with a Pole of 1.80 m as target (continuation of Table 3).

| | DJI143 | DJI144 | DJI148 | DJI149 | DJI150 | DJI151 | DJI152 | DJI153 | DJI154 | DJI155 | DJI156 | DJI157 | DJI40 | DJI43 | DJI137 |
|---|---|---|---|---|---|---|---|---|---|---|---|---|---|---|---|
| Number of pixels upwards ($r_i$) | 207 | 359 | 32 | 83 | 68 | 112 | 241 | 156 | 307 | 339 | 573 | 902 | 254 | 114 | 136 |
| Number of pixels downwards ($r_j$) | 208 | 75 | 86 | 26 | 102 | 90 | 145 | 78 | 304 | 280 | 559 | 211 | 180 | 161 | 33 |
| Gimbal pitch angle (degrees) | 38.5 | 39.4 | 6.9 | 16.6 | 30.4 | 47.3 | 20.7 | 10.7 | 34.4 | 34.4 | 34.5 | 41.7 | 22.9 | 22.4 | 15.5 |
| Flight roll angle (degrees) | 9.6 | 8.5 | 5.5 | 8.6 | 3.1 | 3 | 3.8 | 5.2 | 3.3 | 4.2 | 2.8 | 7.1 | 1.9 | 1.1 | 5.6 |
| Horizontal distance (H) (m) | 9.51 | 9.71 | 54.24 | 54.49 | 28.33 | 15.33 | 14.85 | 26.68 | 7.43 | 7.3 | 4.22 | 4.13 | 13.38 | 21.08 | 37.12 |
| Total number of pixels (after distortion correction) | 415 | 434 | 118 | 109 | 170 | 202 | 386 | 234 | 611 | 619 | 1132 | 1111 | 434 | 275 | 169 |
| Est. height (m) | 1.79 | 1.84 | 1.79 | 1.79 | 1.79 | 1.84 | 1.78 | 1.77 | 1.83 | 1.81 | 1.95 | 1.96 | 1.86 | 1.87 | 1.85 |
| Error (m) | −0.01 | 0.04 | −0.01 | −0.01 | −0.01 | 0.04 | −0.02 | −0.03 | 0.03 | 0.01 | 0.15 | 0.16 | 0.06 | 0.07 | 0.05 |

Positional UAV readings (position of the UAV in WGS84 geographic coordinates) and camera pitch angle of each image were extracted from EXIF tags while the number of pixels spanning upward from image principal point ($r_i$) and downward ($r_j$) were measured manually on screen. The Lat–Long coordinates of each image were plotted into a GIS environment along with the position of the reference pole to measure the horizontal distance (H values, as graphically described in Figure 6).

Besides the description of the relevant parameters used for the calculation, the table reports the height of the target (Est. height), representing the estimated height of the pole considering the horizontal distance obtained with the position indicated in the EXIF tags.

The Root Mean Square Error (RMSE) for the images collected during the Field Test 1 is equal to 0.016 m. On the other hand, the RMSE for the images collected during the Field Test 2 is equal to 0.075 m. In both cases the RMSE was calculated using the error row (real height value—estimated value) for the two data sets and the adequate formula (23), where $x_i$ is the $i$-th value in the row, $x_0$ is the real value of the target height and $n$ is the number of measurements (number of photographs measured).

$$RMSE = \sqrt{\frac{1}{n} \sum (x_i - x_0)^2}. \tag{23}$$

As expected, the *RMSE* of the Field Test 2 is much higher than the other. This is clearly due to the positional uncertainty of the UAV and the target.

This field test has similar conditions as those discussed in Section 2.6: the UAV positioning has an uncertainty of ±1.5 m, the uncertainty associated with the position of the target is ±1.0 m, the uncertainty related to the number of pixels spanning the feature is ±3 pixels at the top and ±3 pixels at the bottom and uncertainty related to pitch is ±1°. Therefore, the equation identified in Figure 9e ($y = 2.7182x^{-0.905}$) can be used to calculate the uncertainty using H. However, as already discussed, only for values above 15 m this can be cautiously used. Figure 12 reports the 10 images extracted from Tables 3 and 4 with H above 15 m and, for each of them, the uncertainty is displayed. These values are also reported graphically in the same figure. The real height of the target (1.80 m) is included in the range of each image.

| Image ID | H (m) | Estimated target height (m) | Uncertainty (m) |
|---|---|---|---|
| DJI151 | 15.3 | 1.77 | 0.23 |
| DJI34 | 17.1 | 1.84 | 0.23 |
| DJI35 | 17.4 | 1.78 | 0.21 |
| DJI43 | 21.1 | 1.83 | 0.21 |
| DJI42 | 21.1 | 1.87 | 0.17 |
| DJI41 | 21.3 | 1.84 | 0.17 |
| DJI153 | 26.7 | 1.83 | 0.17 |
| DJI137 | 37.1 | 1.77 | 0.14 |
| DJI148 | 54.2 | 1.85 | 0.10 |
| DJI149 | 54.5 | 1.79 | 0.07 |

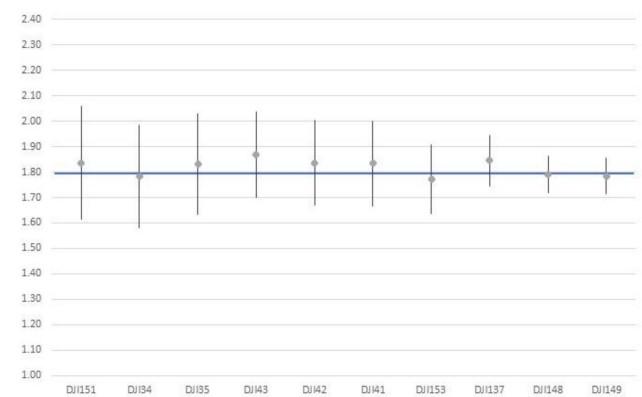

**Figure 12.** Subset of images with H higher than 15 m (see Tables 3 and 4). For each image, the'estimated height and the uncertainty calculated using the function obtained in Figure 9e are reported. The estimated height and uncertainty are also reported graphically for each image (on the right).

## 4. Conclusions

The main purpose of this paper is to define how the height of a target can be swiftly estimated using the pinhole model tailored for gimballed cameras mounted into UAVs. Additionally, it identifies a minimum set of "required" parameters for the calculations and provides a detailed accuracy analysis to define the error associated with the estimation.

The procedure foresees the camera calibration and image distortion compensation before using a pinhole model to calculate geometrically the vertical length of a feature. A pinhole model was chosen for this purpose since it does not need extensive processing nor vanishing lines or reference objects in the scene and, therefore, it is suitable for activities outdoor. The main strengths of this approach are also simplicity and rapidness. Height estimation does not require double cameras, just a single still image acquired with an optical camera is enough. This study analyzed how to adapt the pinhole model for height estimation considering gimballed cameras mounted into UAVs, considering camera vision and UAV flight parameters. Only a few parameters are required: intrinsic camera parameters, which are usually provided by the manufacturer but can also be retrieved via camera calibration, camera pitch angle, usually available in the metadata associated with the acquired images, number of pixels spanning the feature, which can be measured either manually or automatically using feature detection algorithms, and, finally, the distance between the camera and the target. This latter parameter can be obtained by either considering the difference between the UAV and target coordinates or using a laser range finder if the UAV is fitted with such a device. Consumer-level UAVs generally do not have laser range finders. Thus, the coordinates of a moving target, like a human, can be only retrieved if the target is over, or in the immediate proximity of, a recognizable feature, like a building corner or street marker in urban areas.

The processing for lens distortion compensation, which may be time-consuming if performed for the entire image, is applied in a fashion that drastically reduces the processing time because it involves only a very limited number of pixels. In those cases where the counting of the pixels spanning the feature can be performed automatically and the instantaneous slant range measure is available, the estimation of the height can be done in real-time due to the limited computational burden required by this approach.

This paper has also analyzed how the parameters involved in the calculation (horizontal camera-to-target distance and number of pixels spanning the target in the image plane) are affecting the overall uncertainty. The analysis indeed highlighted a fact: the error generated by the combination of uncertainties described above has a well predictable behavior when plotted against the horizontal target-to-camera distance. The effect of the error in H may decrease with increasing H; moreover, as H increases, the number of pixels decreases, and error associated with the counting of the number of pixels increases. The uncertainty analysis has shown that the overall error in the determination of the target height has a flat behavior between 15/25 m and 50 m, while for a greater distance the error may rise again. This indication is clearly true for the conditions considered in this study which are the most probable scenario. Moreover, this was also confirmed by real data collected with the UAV DJI Phantom 4 PRO. Values collected for the validation of the method, both in a controlled environment and in an outdoor environment, were nevertheless much better than the estimated uncertainty. Based on the test field results, we can say that the uncertainty of the presented target height determination method using a DJI Phantom 4 Pro is ±0.075 m. We also underline that DJI Phantom 4 PRO is a valid device but, still, a consumer-level UAV with tangible limitations for surveillance purposes, especially in terms of the positioning system and metadata availability.

In future developments, a more performant UAV (in terms of positional accuracy) should be considered to reduce the uncertainty related to UAV positional error. Moreover, the possibility to use existing or *ad hoc* ground markers in urban environments to enhance the likelihood to indirectly retrieve the position of mobile targets should be further explored.

**Author Contributions:** Conceptualization, A.T.; methodology, A.T.; software, M.C.; formal analysis, A.T.; resources, M.P.; writing—original draft preparation, A.T.; writing—review, M.P., M.C. and P.R.; supervision, M.P., P.R. and M.C.; project administration, M.P. All authors have read and agreed to the published version of the manuscript.

**Funding:** The work was partially supported by FCT through the project UIDB/50019/2020—IDL and national funds through the FCT (Fundação para a Ciência e a Tecnologia) by the projects GADgET (DSAIPA/DS/0022/2018) and AICE (DSAIPA/DS/0113/2019). Mauro Castelli acknowledges the financial support from the Slovenian Research Agency (research core funding no. P5-0410).

**Acknowledgments:** The authors would like to express their gratitude to José Cortes for the support in data acquisition, Marta Lopes and Matias Tonini Lopes for the support in data acquisition and analysis, António Soares for the support in python programming and, finally, Luís Simão, Inês Simão, Ana Martins and Francesca Maria Tonini Lopes for the logistical support.

**Conflicts of Interest:** The authors declare no conflict of interest.

**Disclaimer:** The content of this paper does not necessarily reflect the official opinion of the European Maritime Safety Agency. Responsibility for the information lies entirely with the authors.

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
