# Peer review of "Remote Estimation of Target Height from Unmanned Aerial Vehicle (UAV) Images"

_remotesensing, doi:10.3390/rs12213602_

Round 1

Reviewer 1 Report

The artical is appropriate for publication in present form.

Author Response

“REMOTE ESTIMATION OF TARGET HEIGHT FROM UNMANNED AERIAL VEHICLES (UAVs) IMAGES”

Andrea Tonini, Paula Redweik, Marco Painho and Mauro Castelli

Response to Reviewer 1 Comment

The Authors would like to thank the Reviewer 1 for the revision of the paper and the comments provided.

Reviewer 2 Report

The authors present a practical concept, however, no guidance is provided on the limitation of this work and I am not sure if this has been validated thoroughly. The one big assumptions in this work states that the "distance between the camera and the target" is known.  The authors speculate that this can be obtained using a laser range finders in the UAS, however, this is not demonstrated. 

With that major assumption aside, no other limitation is provided in terms of altitude, distance, camera angle, etc.  Also, an estimated error of 7.5 cm is good, but the fact that it is not a function of the pixel size is quite surprising. 

This assumption is key, because without it, the scale is only known relatively.  For this reason, I am not sure how this work can be actually deployed outside of a controlled environment.  For this reason, and in the opinion of this reviewer, I am sending it back for additional review/consideration. 

The grammar and formatting of the manuscript should also be carefully reviewed.  

Reviewer 3 Report

Abstract

27-28 Your application is not rapid: you need to select the image, to find a specific “target” and point to it, find the same “target” in your GIS: relating the object and the image isn’t straightforward. The remaining steps might involve a few clicks only, I agree, but overall there is little automation if you need GIS data.

Introduction

71-72 that UAVs are capable to support image metrology is not a novelty; unless you show that you need to overcome a specific problem with the new platform, just applying the same technique to different platforms is not a new approach.  

71-72 a few lines above this sentence, you said that machine learning is time consuming; so I don’t see why it’s becoming attractive in your case now.

86 maybe “computed” is more appropriate than “identified” here

91 I would drop “exact”: measurements always contain errors

105-105 “… it is analysed how computer vision can be performed…” you may apply a computer vision technique, not performing computer vision… please rephrase

199 In the paper referenced [21] I couldn’t find any statement about the a “very good” relative accuracy of gyroscopes. Please check and in case correct. In any case, relative measurement might be good, however it’s well known that drift is the problem with inertial measurements.

208-213 I don’t see why the target should be centred in the image plane (“because the method is meant for outdoor activities where the topography may rapidly change”). Please explain.

227 2.4 Computer vision with cameras installed into UAVs

I don’t understand the title of this paragraph. The main topic is gimbal and camera orientation that has nothing to do with computer vision. Your method (at least in this paper) has no automation… Maybe you should call this simply On-board Sensor orientation as you are simply describing how to use on-board sensors to compute camera exterior orientation elements: this has more to do with navigation or direct sensor orientation.

230-247 in describing the sequence of transformations form [25] you don’t give the reader enough details… though I understand this is not a “core business” of the paper. I would not use “Inertial frame” to name what is in fact the local level frame; moreover, the Object reference system here is missing unless you assume it to coincide with the local level frame.

262-266 The accuracy of 0.5 m and 1.5 m you mention refer not to positioning but to hovering in the Dji manual. I’m not sure you can assume them to be the same. As far as GPS is concerned, it is against all empirical and theoretical studies that vertical accuracy is better than horizontal accuracy. The angular accuracy refers to the gimbal zero reference, not to the pitch.

Figure 4 (a) is in fact figure 5 (a) and vice versa (roll=0 and roll =/ 0)

342-343 I don’t understand why you say “H and V are not related at all to the topography” (and why this should be important) … H and V depend on the relative height and horizontal distance between UAV and target foot. If instead you want to say that you need only the target horizontal position and not its elevation, ok, just say that.

382-386 When you say that, with moving targets, the method as it is doesn’t work, I guess this is because the target is not in the image centre. Please explain why you cannot slightly modify your method to accommodate for “off-centre” targets.

Figure 8: Image Extrinsic params include Pitch and Camera position, so the information flow should be rearranged: the box position is in a wrong position.

2.7 Your error source analysis needs improvements: first of all the proper statistical tool to this aim is the error propagation law; second, you are disregarding the object position precision (GIS data normally are measured photogrammetrically, not by a GPS pole, so errors depend on the nominal scale of GIS data and range from several dm to more than 1 m); precision of collimation at top and bottom of the pole: the further you go from the object, the more this error will show up (at 50 m 1 pixel error is 1.5 cm with your data, and people’s head is not as easy to pinpoint as the top of a pole…); most important: I doubt that the 0.02° angular accuracy refer to the drone body yaw pitch roll: this would be a good value for a medium accuracy INS (not on board of low cost UAVs); in the specifications this value apply to the gimbal. As you need the camera body pitch (with respect to the local level frame), your pitch data is not as accurate as you assume. Should the P4 camera orientation angles be known to 0.02°, the P4 RTK version could conceivably apply direct georeferencing in the short range… but Dji doesn’t claim this.

Figure 9: Rhinoceros is not necessary to get this result. A computation in excel is enough and would be more versatile.

435-442 using error law propagation would show this straightforwardly, without running 20 cases with random positions. I don’t think selecting positions at random is the way to get hints on the error behaviour: you must change a single parameter each time.

Table 1: please fill the table for increasing values of one parameter (H, Pitch…), not in random order. (and don’t use randomly generated (H, Pitch) pairs).

451-461 this paragraph needs improvements, especially as far as points b) and c) are concerned

Figure 10 and 11 do not make sense in my opinion, especially including a fit and coefficient of determination: this is not a scatter plot where scattering depends on random errors. In both cases you put in a mix of H and Pitch values, while if you want to have the picture of both dependencies at once you’d better plot several data point sets, each with a constant value of H or Pitch. The total n. of pixel is not an independent parameter in your simulation, so in fact you are not decoupling it from the other parameters. As far as Figure 12 is concerned, it is true that if you compute the effect of the uncertainty in H, the error increases as H decreases and that the error is almost insensitive to pitch values. That’s why your 20 random cases fit nicely on a curve. However, if you try with a higher target (e.g. a building) you’ll find that for large pitch values the effect is noticeable. Finally, why are the effects of errors in pitch and image measurement of top and bottom of the target being ignored? they might be negligible, but you have to demonstrate it.

542-544 It’s not clear what is the Distortion coefficient mentioned here. Please clarify.

r 655 Also, the processing for lens distortion compensation, which may be time-consuming if performed for the entire image, is presented in a novel fashion that drastically reduces the processing time because it involves only two pixels

This is not a real novelty: in analytical photogrammetry this has been done for more than a decade…

669-671 Measuring tree height is more complex than you imagine, your method would not work well: the tree top (however you define it) is not necessarily on the vertical of the tree foot.

Round 2

Reviewer 3 Report

As a general remark, I would read again the Introduction, as part of the newly added text does not fit well with the previous… Look especially to cases where successive statements often repeat basically the same concept.

Moreover, try to explain more clearly what is your proposed method for: you say it’s basically to measure people’s height, in real time, in a range 10-20 m (not much more), as far as I understand. How does it square with regulations on critical operations of UAVs? Why is measurement needed in real time?  

Check also references, some are incomplete (e.g. [16])

59 machine learning approaches are performed to images acquired… better: machine learning approaches are applied to images acquired…

101  estimation. and reinforce   … delete the ‘.’

 208-211 The “perfect” approximation would be a polynomial of infinite degree; however, this precision is not needed. Several studies, such as [17], confirmed that for average camera lenses the first order is enough, while more coefficients are required for fish-eye lenses

Sorry, this point needs clarifications….

  1. a) I can list you many studies from photogrammetry that demonstrate quite the opposite, also for “average camera lenses” and specifically with UAV lenses. Everything depends on the accuracy you need in modelling distortion (1 pixel down to 0.1 or 0.05 pixels): in photogrammetry you need to be very accurate and account for other distortion types, e.g. lens decentring. Your statement above gives the impression that the odd polynomial model can be used also for fish eye, which is totally wrong. Search internet for distortion models for fish eye and you’ll see why.
  2. b) It’s not clear to the reader what “first order” means. Maybe you mean the first coefficient k1…
  3. c) Be aware that some manufacturer (Dji among them) apply distortion correction by image resampling according to a pre-defined correction model (have a look to Dji manual) in the on-board camera, so the images you process are already corrected for the largest effects of radial distortion. This, and the fact that your collimations are close to image centre, explain why your corrections are negligible (see Table 3 first submission). This fact should be mentioned in the experimental section, if applies.

So I suggest to modify the sentence saying “In the light of our error propagation analysis (see Section …) … “and specify your correction goal (half pixel?) and show why (if so) the k1 coefficient is enough.

Please remove reference to fish eye lenses.

237-238 Intrinsic parameters can also be obtained from a bundle adjustment of a regular photo coverage with high image overlapping.

Rather than “high image overlapping “ better “highly convergent images”. A reference is needed here. I recommend

Remondino, F., & Fraser, C. (2006). Digital camera calibration methods: considerations and comparisons. International Archives of the Photogrammetry, Remote Sensing and Spatial Information Sciences, 36(5), 266-272.

Figure 4 (a) and Figure 5 (a) have been mistaken: look in comparison to both and you see that Figure 4 (a) depicts in fact the case with roll different from zero while Figure 5 (a) depicts the case with roll different from zero. (the drawings (b) are correct, instead.)

281 Angular values are measured with an uncertainty of ± 0.02° [23].

This is a key point of the paper in my view. As I already note in the comments to first submission, the angular accuracy refers to the gimbal zero reference, not to the pitch. This is clear from a simple reasoning:

should the 0.02° angular accuracy refer to the drone body, so it should as well apply to yaw and roll (in IMU usually pitch and roll are a bit better than yaw): this would be a good value for a medium accuracy INS (not the type on board of low cost UAVs. As you need the camera body pitch to add to the gimbal angular value, your pitch data is not as accurate as you assume. Should the P4 camera orientation angles be known to 0.02°, the P4 RTK version could conceivably apply direct georeferencing in the short range… but Dji doesn’t claim this. The fact that your test data are fine just applying the pitch values registered by the P4 doesn’t mean the user can expect this to be true in general.

403-418 What you state in this section makes it clear to me that your system is not designed for the P4 UAV-class (micro-UAVs) and that even the claimed possible improvements mentioned (automatic target localization) will not solve the problem i.e. the measurement of UAV-target distance. Again, nothing will make your system real-time and its practicality (using Google Earth and building corners) is really disputable. I also find unconvincing your answer to my remarks on first paper version (previous row 382-386).

2.6

422-426 I disagree with this statement about angular measurement (see also above). The pitch value you get from the Exif is not accurate to 0.02°, so you should also propagate this error. I made a few tests on Table 1 data from 0.1 to 1° error. For 20 m distance and 30° pitch it goes from 3.5 mm to 3.5 cm. It’s likely to be linear for any given combination pitch-H and it’s quite strong for high pitch and short H.

 481 This analysis tells us that, statistically, we should expect an error of ± 0.04m

please recall that this is true at 20 m and 30° pitch (you may stress this is your ideal operating case, if so).

497 the reciprocal errors… better the correspondent errors

Figure 9  about e) plot. It’s true that for H increasing the effect of error on H decreases, however the error due to pix counting increases with increasing H. According to my computations (see also remark on row 702-707) the combined effect of error in H and pix_counting decreases with a flat min at 5.4 cm around 40 m then starts growing again (as common sense would have). From your graph perhaps it is also so, the last dot near H=50 m looks above the one at 45 m… Is it so?

598-600 The Distortion Coefficient to be used for the correction was retrieved through camera calibration techniques [15] developed with OpenCV via Python programming and is equal to 0.0275.

Two remarks: first, simply say whether this is k1 coefficient in the model (12); second, the value 0.0275 in itself says nothing to the reader, that need not to be familiar with the order of magnitude, so a comment is necessary.

702-707 The analysis indeed highlighted a very interesting fact: the error generated by the

combination of uncertainties described above has a well predictable behaviour when plotted against

the horizontal target-to-camera distance. The uncertainty related to the height is generally lower at

higher distances and it grows, with a negative power function, when the UAV gets closer to the target,

independently from the pitch angle and the number of pixels spanning the feature.

A few remarks on this point. 1) I would not use “interesting fact”: that a trend estimated by error propagation law provides a predictable behaviour is the basis for survey design and in general one of the reason for using this law, so nothing new. 2) that the effect of the error in H may decrease with increasing H is quite acceptable. What is wrong is the way you write the statement, as it leads the reader to think that this is the overall accuracy (and so that the farther the UAV moves from the target, the better his measurement accuracy will be, which is plainly absurd). In fact, as H increases, the number of pixels decreases and the collimation error effect on estimation of height increases. I made a simulation with V in the range 16-22 m and H from 15 to 50 m (h=1.8 m), error in H= 0.75 m, error in counting 3 pixel (overall). While the error due to H decreases from 8 to 3 cm, the error due to pixel counting grows from 2.5 to 4.8 cm.

So please rephrase the two statements.

713-714 Better UAVs will offer better performance and therefore more accurate estimation of target height.

I beg to disagree with this statement. It’s not P4 that has limitations, it simply that you got an idea that requires an on-board range finder that manufacturers don’t find necessary in most applications. As long as there is no range finder, your method will simply not work in practice according to your goals (speedy, almost real-time) and accuracy won’t be in a range that might be useful e.g. to claim “this people is likely to be that person”. There is already on the market the P4 RTK version that provides cm-level positioning accuracy, but this solves only the easier half of your problem.
